# Electrosprayed Alginate Nanoparticles as CRISPR Plasmid DNA Delivery Carrier: Preparation, Optimization, and Characterization

**DOI:** 10.3390/ph13080158

**Published:** 2020-07-22

**Authors:** Batoul Alallam, Sara Altahhan, Muhammad Taher, Mohd Hamzah Mohd Nasir, Abd Almonem Doolaanea

**Affiliations:** 1Department of Pharmaceutical Technology, Kulliyyah of Pharmacy, International Islamic University Malaysia, Kuantan 25200, Malaysia; alallamb@gmail.com (B.A.); mtaher@iium.edu.my (M.T.); 2College of Pharmacy, Alfaisal University, Riyadh 11533, Kingdom of Saudi Arabia; saltahhan@alfaisal.edu; 3Department of Biotechnology, Kulliyyah of Science, International Islamic University Malaysia, Kuantan 25200, Malaysia; hamfizah@gmail.com; 4IKOP Sdn Bhd, Kulliyyah of Pharmacy, International Islamic University Malaysia, Kuantan 25200, Malaysia

**Keywords:** clustered regularly interspaced short palindromic repeats (CRISPR), CRISPR-associated protein (Cas9), alginate, gene delivery, transfection, nanoparticles

## Abstract

Therapeutic gene editing is becoming more feasible with the emergence of the Clustered Regularly Interspaced Short Palindromic Repeats (CRISPR)/CRISPR-associated protein (Cas) system. However, the successful implementation of CRISPR/Cas9-based therapeutics requires a safe and efficient in vivo delivery of the CRISPR components, which remains challenging. This study presents successful preparation, optimization, and characterization of alginate nanoparticles (ALG NPs), loaded with two CRISPR plasmids, using electrospray technique. The aim of this delivery system is to edit a target gene in another plasmid (green fluorescent protein (GFP)). The effect of formulation and process variables were evaluated. CRISPR ALG NPs showed mean size and zeta potential of 228 nm and −4.42 mV, respectively. Over 99.0% encapsulation efficiency was achieved while preserving payload integrity. The presence of CRISPR plasmids in the ALG NPs was confirmed by Attenuated Total Reflectance-Fourier Transform Infrared Spectroscopy. The tests revealed that the nanoparticles were cytocompatible and successfully introduced the Cas9 transgene in HepG2 cells. Nanoparticles-transfected HepG2 was able to edit its target plasmid by introducing double-strand break (DSB) in GFP gene, indicating the bioactivity of CRISPR plasmids encapsulated in alginate nanoparticles. This suggests that this method is suitable for biomedical application in vitro or ex vivo. Future investigation of theses nanoparticles might result in nanocarrier suitable for in vivo delivery of CRISPR/Cas9 system.

## 1. Introduction

In the past decade, gene editing Clustered Regularly Interspaced Short Palindromic Repeats (CRISPR)/CRISPR-associated (Cas) systems have become a platform in biomedical research due to its simplicity, precision, and versatility [1,2]. CRISPR potential has been recently explored in gene therapy and has shown efficient gene editing activity for many therapeutic purposes [3]. Despite the successful use of CRISPR technology in the precise in vitro genetic engineering, in vivo applications are still lacking and newly emerging. The successful clinical application of genetic material-based strategies remains dependent on the generation of safe and effective delivery systems that have the ability to overcome numerous biological barriers associated with delivery of plasmid DNA into target cells. Intracellular delivery of naked plasmid DNA can be a challenge as it cannot cross through the cell membrane due to its large size, net negative charge, and hydrophilic property. In addition, naked plasmid DNA is very sensitive to degradation in the body especially nuclease enzyme. Exposure to plasma enzymes can destroy the plasmid. Therefore, delivery vehicles are often employed for easier passage across membrane while preserving the plasmid from enzymatic degradation [4]. Till now, the large majority of CRISPR plasmid therapy have been based on the use of viral vectors, like adeno-associated virus (AVV) [5,6], lentivirus (LV) [7], and adenovirus (AdV) [8]. Although viral vectors exhibited high gene delivery and expression efficiency [9], these carriers are generally concerned with safety issues as their potential in immunogenicity [8], carcinogenesis [10], and limited packaging size [11,12].

The non-viral delivery systems for CRISPR system are a safer alternative to viral vectors which offer more flexibility in terms of the size of the genetic material they can deliver and carry low risk of immune responses commonly associated with viral delivery [1]. Nevertheless, its delivery efficiency is lower than viral carriers. Non-viral vectors are classified into physical and chemical delivery method methods. Physical delivery methods, like microinjection [13], electroporation [14], nucleofection [15], and hydrodynamic injection [16] have been reported to deliver CRISPR/Cas9 system in vitro [17] and in vivo [18]. The delivery of the genetic material in these methods is facilitated by a physical force which opens transient nanometer-sized pores within the cellular membrane to allow genetic material flow into the cytoplasm or, in some cases, the nucleus. Even though high efficiencies can be achieved, a major disadvantage of these physical methods is they are not suitable for in vivo use due to their technical challenges and variable physiological and pathological conditions [1,19].

The chemical delivery methods are divided into four groups based on material of delivery used for the nanoparticles (NPs), and they can be classified as lipid-based, polymeric-based, lipid-polymeric hybrid, and inorganic nanocarrier [20]. Lipid-based gene transfer was one of the earliest strategies used in gene therapy. The most frequently used cationic lipid-based vectors include liposomes and solid lipid nanoparticles. Some of the cationic lipid-based delivery systems, such as Lipofectamine (cationic lipid), are commercially available through transfection reagents and have been used to deliver CRISPR nucleic acids [21,22]. Despite the ease of use and convenience of commercial transfection reagents, the main drawbacks of cationic lipid-based nanocarriers were the limitations of carrying large size Cas9/guide RNA (gRNA) plasmids and insufficient shielding of the plasmid’s negative surface charge [23], in addition to their poor stability [20].

Polymeric vectors were later developed to overcome the problem associated with lipid-based nanocarriers. Delivering a CRISPR/Cas9 system using polymeric nanoparticles has been attempted with variable success rates. Initially, synthetic polymers were used. Polymeric vectors, such as Polyethylenimine (PEI) [24,25], and Polyamidoamine (PAMAM) dendrimers [26] were used to deliver CRISPR plasmids. Since they are positively charged, they can condense the negatively charged nucleic acids, thus forming small and enzymatically stable polyplexes. Nevertheless, maintaining the balance between transfection efficiency and toxicity has been the major concern during the design and preparation of positively charged carriers. As the transfection efficacy and toxicity were strongly correlated with the molecular weight and structures, the higher molar mass and higher positive charge, the higher condensing action with nucleic acids. Nevertheless, this is also accompanied with markable increase in toxicity. Thus, low transfection efficiency and cytotoxicity associated with these polymers, in general, are considered major limitations for the in vivo application [27].

As a consequence, the type of nanoparticles that received much consideration was fabricated from poly (lactic-co-glycolic acid) (PLGA) [28,29]. PLGA is considered as one of the most successfully developed biodegradable polymer developed, since its degradation leads to lactic acid and glycolic acid metabolite monomers; thus, a minimal systemic toxicity is associated with PLGA usage [30]. Despite the interesting results for PLGA nanoparticles reported in literature, this system is also concerned with limitation, mainly because of the transfection efficiency [31]. Therefore, the need of the ideal carrier can be satisfied by using natural polymers, which are derived from the biological systems, like plants, animals, and microorganisms [32], because of their abundant nature; they can be obtained through the renewable processes due to their sustainability, biodegradability, and biosafety [33,34]. In addition, due to their tunability, several degrees of anionic, cationic charge, and cross-linking can be added during polymer production via controlling the reaction conditions, thus changing their capabilities as a nucleic acid delivery vehicle [35]. This makes them a promising vehicle for efficient CRISPR/Cas system delivery.

Sodium alginate (ALG) is a linear unbranched anionic natural polysaccharide composed of two monomers of uronic acids linked via (1–4) glycoside linkage, *α*-l-guluronic acid (G), and β-d-mannuronic acid (M). The residues are arranged in three ways of blocks along the chain: M blocks, G blocks, and blocks of alternating G-M residues. The sources from which alginate is isolated will determine the amount and distribution of each unit. Alginates from brown seaweed sources show different chemical and structural properties based on their seasonal and growth conditions [35]. The relative amounts of these two monomers (M and G units) strongly influence total chemistry of the polymer, such as transmittancy, swelling, and viscoelasticity, while the ratio of G/M blocks determines its permeability [36]. The chain blocks flexibility decreased along the series GG > MM > MG [35,37]. Hence, G content in alginate affects the mechanical strength and physical stability of the gel. In addition, the physical properties of alginate polymer are also influenced by the concentration of the cross-linking cation used [38]. Therefore, selection of material molecular weight, G/M ratio, and cross-linking ion will affect final gel properties, as more homogeneous gel without porosity can result from high content guluronic acid alginate with a high affinity cations for alginate [39]. Moreover, alginate biopolymer is considered an optimal selection for microencapsulation of bioactive ingredients due to its biocompatibility, biodegradability, and mucoadhesive properties [40]. Moreover, its natural origin, low price, availability, and sol-gel transition properties make alginate an ideal candidate to produce particles for different applications. Formation of alginate hydrogel is simple, mild, stable, and eco-friendly, with the main function of entrapping active molecules rapidly [41]. Several techniques have been developed and optimized to prepare the alginate micro-and nano-particles, including complexation, solvent evaporation/diffusion, emulsification, spray drying, and solvent-casting methods; nevertheless, the particles produced were often highly porous, and they were difficult to scale and reproduce. Moreover, the use of temperature or volatile organic solvents has also been a limitation for these methods [42].

Recently, preparation of polymeric particles using electrohydrodynamic atomization (electrospray) technique has received a great interest [43]. Compared with the conventional methods to prepare nanoparticles, electrospray has many advantages. Electrospray method is a one-step technique, with the potential to control the size and polydispersity of the particles while preserving the activity of the payload in a reproducible manner [44,45]. It is scalable and can be easily automated to form small to large scale production. This makes it an ideal technique for production of nanoparticles. Basically, electrospray can be implemented by applying a sufficient high electrical field to polymeric solution infused through a capillary nozzle to force the polymer to come out of the nozzle and create nano- to micro-droplets [44]. Despite the simplicity and quickness of such an encapsulating system, the surface tension of the polymeric solution is a limiting step for small monodispersed homogeneous particles production. This issue could be overcome by proper selection of the solvent and careful adjusting of formulation and process variables [46].

Electrospray of aqueous alginate solution was often reported as either the pulsating jet model or dripping model. Consequently, alginate beads or microspheres are often formed. The generation of nanoparticles from alginate solution requires the work under inert gas environment (e.g., CO_2_ or Ar) or vacuum owning to the high surface tension of the solution. In addition, encapsulating hydrophilic materials in alginate nanoparticles is challenging with the diffusion of the payload into the gelling medium due to the high porosity nature of alginate particles. One good example here is encapsulation of whey protein, in which the encapsulation efficiency (EE) achieved was just below 60% [47]. The major challenges associated with plasmid delivery, via polymeric carrier, were the limitations of carrying large size plasmids, hence insufficient shielding of the plasmid’s negative surface charge [23] and, consequently, low transfection efficiency. Therefore, this study aimed to develop ALG nanoparticles using electrospray technique and evaluate the potential of utilizing them for gene delivery. Two CRISPR plasmid were used as models.

Herein, we present the generation of biocompatible alginate nanoparticles (ALG NPs). The effect of formulation characterstics (surface tension, viscosity, and electrical conductivity) was investigated as they highly affect the cone jet formation, hence the dispersability of the particles. Moreover, the process variables (flow rate and applied voltage) were optimized to achieve the smallest particles size, since particle size is a key element for gene delivery application. Then, the particles were tested in HepG2 cells for cytotoxicity and transfection efficiency. Additionaly, primary results of gene editing capability were also shown as a proof of concept, which may require further optimization.

## 2. Results and Discussion

### 2.1. Encapsulation of CRISPR Plasmids in ALG NPs

#### 2.1.1. Optimization of Formulation Variables

Encapsulation of CRISPR plasmids (pDNA) in ALG NPs was conducted using electrospray technique as shown in Figure 1. Prior to that, experiments were conducted to determine the ALG solution (S) properties that can form a stable cone jet. Previous studies reported that stable cone jet in the electrospray is crucial to form particles with narrow size distribution in a reproducible manner [44,45]. Therefore, the physiochemical properties of eight alginate solutions containing sodium alginate alone or with Tween 20 were tested, as presented in Table 1. As sodium alginate concentration increased, electrical conductivity, viscosity, and surface tension increased significantly; however, there was a slight increase in pH and density. All ALG solutions exhibited Newtonian behavior, whereas S6 exhibited non-Newtonian shear thinning behavior (Figure 2).

Even though solutions with ALG concentrations less than 1% *w/v* (S1 and S2) were able to form spray, the spray was not stable. This can be explained by the fact that the successful of stable cone jet and the breakup of droplets at the nozzle tip are related to physiochemical properties of electrosprayed solution, among these, surface tension. High surface tension liquids trigger coronal discharge at nozzle tip, which in turn changes the stable cone jet into an irregular spraying and an asymmetrical mode since it prevents atomization of charged particle [48]. According to Smith (1986), a liquid with surface tension value higher than 50 mN/m cannot be atomized in air by electric forces [49]. Electrospraying high surface tension aqueous solution usually require the work under inert gas environment or vacuum. Hence, all ALG solution surface tension was higher than 50 mN/m; however, as S1 and S2 had a significantly lower surface tension compared to other ALG solutions (S1 to S6), they were still able to form an unstable cone jet.

Furthermore, all solutions which have ALG concentrations more than 1% *w/v* (S4 to S6) were too viscous to be extruded from the nozzle tip. In this case, the viscosity of ALG solutions prevents the formation of stable cone jet, as the drying of the particles occludes the nozzle tip, therefore limiting the stability of the Taylor cone [50]. Thus, there is an appropriate range of viscosity for achieving cone-jet mode during the electrospraying process.

Interestingly, the electrical conductivity also plays an important role in the cone jet formation as it affects the charge quantities on droplet surface and further influences the strength of electric forces. Several researchers have concluded that cone jet was obtained only if the electrical conductivity was set in certain range [48]. Drozin concluded from his research that liquids of conductivity higher than 10^−12^ S/m cannot be sprayed by the electrohydrodynamic method [51]. Electrospraying in the cone-jet mode is possible when the liquid conductivity falls within the range of 10^−5^–10^−11^ S/m [52], or 10^−1^–10^−11^ S/m [49]. Nevertheless, each liquid has more specific range.

A stable electrospray process was achieved when Tween 20 was added to the alginate solution (S7 and S8), as shown in Figure 3. Tween 20 significantly reduced the surface tension of the ALG solution leading to a stable cone jet formation. It can be seen that the surface tension dropped almost to half, from 65.02 ± 0.01 mN/m in S2 to 36.29 ± 0.16 mN/m, in S8, after Tween 20 addition. Tween 20 acts as a non-ionic surfactant that reduces the surface tension without affecting the solution conductivity in contrary to the ionic surfactants that may result in less stability in electrosprayability [53]. As for S7, the solution was also able to form stable spray, as its surface tension was almost similar to S8. However, NPs formed from S7 were not stiff because of lower alginate concentration hence less ion sites for cross linkage. Consequently, the 1% *v/v* Tween 20 in 1% *w/v* ALG solution was selected as the solution to be further used in NP preparation.

#### 2.1.2. Selection of CRISPR pDNA: ALG Ratio

More than 99% encapsulation efficiency (EE) was obtained by using 1:100 and 10:100 pDNA:ALG (w:w) ratio. However, increasing the ratio more than that resulted in decreasing the EE, as it dropped to 92.19 ± 0.29% for 20:100 pDNA:ALG ratio. Therefore, 10:100 pDNA/ALG ratio was selected to prepare the nanoparticles (Appendix A).

#### 2.1.3. Optimization of Process Variables

The 30G nozzle gauge was selected to perform this study as it produced the smallest droplets. According to the reviewed literature, increasing the nozzle diameter increased the particle size. The droplets is typically about twice the jet diameter and by increasing the nozzle size, the jet dimensions will be increased [54]. For instance, Nikoo and co-workers achieved a smaller alginate droplet using smaller nozzle [44]. Similarly, Yaghoobi and colleagues conclude that reducing nozzle diameter leads to decrease in size for electrosprayed PLGA NPs [55]. On the other hand, 4 cm was selected as spraying distance, as distances lower than that resulted in film formation on top of collector in a preliminary study. Distance more than 4 cm was not much preferred as it resulted in reduced yield due to the loss to the surrounding environment during the droplet flight. Longer distance also required higher voltage to achieve the spray. Furthermore, 1.5% *w/v* was selected to be the working calcium chloride concentration. Niko and colleagues had reported that the mean diameter of electrosprayed ALG particles in calcium chloride solution decreases with increasing calcium chloride concentration up to 2% *w/v*. After that concentration, the change was not significant [44]. Additionally, 15 min was selected for curing time, based on preliminary study (no significant difference in encapsulation efficiencies was reported between 15 and 60 min). Therefore, the formation of continuous stable spray is function of selection the appropriate flow rate and applied voltage ranges. Three voltage levels (9.5, 11, and 12.5 kV) were selected, and their influence on median diameter (Di 50) of the particles were investigated using three different flow rates (0.1, 0.3, 0.5 mL/h), while holding the other parameter constant.

Nanoparticle size was selected as the criteria to optimize the preparation because of its critical effect on the NP functions including cytotoxicity [56], cell uptake [57,58], transfection efficiency [59], and biodistribution [60]. Moreover, the EE was close to 100% for all formulations, so it was not selected as a response for optimization. Statistical analysis of the factorial design revealed that the model was significant (*p* < 0.05). The R^2^ of the model was 0.9752, indicating the capability of the model to predict the response (Appendix A).

##### The Effect of Applied Voltage on Particles Size

The main effect plot showed that the mean particle size decreased from 679 nm to 260 nm when the voltage increased from 9.5 to 12.5 kV (Figure 4A). Nevertheless, the interaction plot revealed that the flow rate affected the rate of decrease in the particle size when the voltage increased (Figure 4B). As the voltage increased from 9.5 to 12.5 kV, there was 68.9% decrease in particle size for the flow 0.1 mL/h. However, the particles decreased 56.2% and 58.6% at the 0.3 and 0.5 mL/h, respectively. This particle size reduction is related to the fact that applied voltage to an electrosprayed solution creates charges at the nozzle tip. At a lower applied voltage, only limited charges are on the drop. Increasing the voltage leads to increase in the charge, which increases coulombic repulsion forces (stretching force on the jet segment). Increasing coulombic forces provides more repulsion between adjacent droplets. This leads to smaller particles [61]. This result is in agreement with a study reported by Abyadeh et al. (2017), who mentioned that increasing voltage from 10 to 15 kV resulted in decreasing of diameter of electrosprayed chitosan NPs by 20% [43]. A similar relationship was also reported by Musaei et al. (2017), where the size of poly(lactic-co-glycolic acid) (PLGA) particles decreased by 13.95% when voltage was increased from 8 to 11 kV [62]. Similarly, result from Abyadeh et al. (2019), the model indicates that chitosan/pDNA particles size decrease from 1500 nm to 420 nm after increasing applied voltage from 9 to 12 kV [63].

##### The Effect of Flow Rate on Particles Size

The main effect plot of flow rate on the size of NPs showed that the mean particle size remained almost constant, 533 nm and 531 nm, when the flow rate was increased from 0.1 to 0.3 mL/h. However, further increase in flow rate to 0.5 mL/h resulted in reduction in the mean particle size to 463 nm. Moreover, the percentage of particle size change varied based on the applied voltage. The particle size decreased by 23.2% and 6.8% for the 9.5 and 11 kV, respectively. Therefore, the lower the voltage, the higher decrease in the particle size was. This finding disagrees with the well-known fact that higher flow rate increases the particle size as the polarization time of the droplet is shorter due to faster movement of the droplets toward the collector [48]. However, at voltage 12.5 kV, increasing flow rate resulted in slight increase of 2.4% in the particle size, which is in agreement with previous reports. For instance, Songsurang et al. (2011) concluded that the size of Doxorubicin-Chitosan-Tripolyphosphate NPs increased approximately two- and four-fold, when the flow rate was increased from 0.5 to 5 mL/h and then to 10 mL/h, respectively [43]. Similarly, Xu, Skotak, and Hanna demonstrated that the average diameter of polylactide microparticles increased by 44% at the voltage of 12.5 kV as the flow rate increased from 0.5 to 3 mL/h [64].

The main and interaction effects were used by the Minitab software (version 19) to generate the optimized formulation considering the range of acceptance in response prediction as 95% confidence interval (192.4–280.3) and prediction interval (148.5–324.2) with 0.933 composite desirability. The suggested operation parameters generated from the model were 0.1 mL/h flow rate and 12.5 kV applied voltage (the lowest flow rate combined with the highest voltage used in this study). Consequently, these parameters were used to prepare ALG NPs loaded with CRISPR pDNA (CRISPR ALG NPs) for further investigations.

### 2.2. Characterization of Optimized CRISPR ALG NPs

#### 2.2.1. Particle Size and Zeta Potential Analysis

Figure 4C shows that loading two negatively charged CRISPR plasmids into ALG NPs significantly decreased the mean particle size from 310 ± 5 nm to 228 ± 27 nm and increased the poly dispersity index (PDI) from 0.173 ± 0.016 to 0.428 ± 0.045. This could be attributed to more columbic repulsion forces due to high negative charge of pDNA, which leads to smaller particles. PDI of the particles is a very important factor, defining the dispersity of the nanoparticle size distribution. Most researchers recognize PDI values ≤ 0.3 as optimum values; nevertheless, values ≤ 0.5 are also acceptable [65]. Moreover, loading the pDNA increased the surface charge of the NPs. Zeta potential measurements showed a slightly anionic (−4.42 ± 0.38 mV) surface charge for the CRISPR ALG NPs; the surface was more anionic than blank ALG NPs (−3.37 ± 0.54 mV). Furthermore, more reduction in zeta potential was reported for the particles in the presence of 10% FBS (−5.07 ± 0.62 mV). This reduction could be related to nonspecific adsorption of serum component on the surface of NPs [66].

#### 2.2.2. Transmission Electron Microscopy (TEM)

The morphology of the nanostructure revealed relatively smooth and spherical structure with distinct boundaries, (Figure 4D). In fact, the ultimate morphology of electrosprayed particle was often significantly dependent on the concentration of the polymeric solution. As the concentration decreased, the damage to particle morphology could be increased (particles will appear more porous and fragmented). However, too high concentration could allow fiber structure to occur [67]. Moreover, TEM image confirmed the size and polydispersity of CRISPR ALG NPs, whereby the mean diameter and size distribution appeared to be similar with the dynamic light scattering result.

#### 2.2.3. Attenuated Total Reflectance-Fourier Transform Infrared Spectroscopy (ATR-FTIR)

ATR-FTIR was adopted to investigate the potential interactions in the nanoparticles. The spectra of NPs and the ingredients are shown in Figure 5. Sodium alginate showed a broad band at 3200–3700 cm^−1^ centered at about 3240 cm^−1^ attributed to hydroxyl groups (–OH) stretching vibrations, while the bands at 1591 cm^−1^ and 1415 cm^−1^ corresponded to asymmetric and symmetric carboxyl group stretching vibration, respectively. In addition, the peak at 1080 cm^−1^ was assigned to (C–O) stretching, while the peak at 1022 cm^−1^ was attributed to the glycoside bonds stretching [68]. In the spectrum of Tween 20, the broad band in the range of 3200–3700 cm^−1^ centered at 3477 cm^−1^ attributed to hydroxyl group stretching. Bands at 2921 cm^−1^ and 2861 cm^−1^ correspond to –CH_3_ stretching and -CH_2_ stretching, respectively. Exhibited bands at 1734 cm^−1^ and 1095 cm^−1^ were attributed to the ester (C=O) and (C–O–C) glycoside bond stretching, respectively [69]. In the spectrum of CRISPR pDNA, a broad band centered at about 3477 cm^−1^ attributed to hydroxyl groups (–OH) stretching vibrations, whereas the peak at 1644 cm^−1^ is associated with amine groups present in the nitrogenous DNA bases [70,71]. The characteristic bands at 1555 cm^−1^ and 1409 cm^−1^ come from the stretching vibrations of pyrimidine in DNA [72]. The bands at 1087 cm^−1^ and 1051 cm^−1^ are typically assigned to the vibration of ribose (C–C sugar) [71] or associated with P–O/C–O stretches and PO_2_ symmetric stretches originating from the phosphodiester backbone [73]. The band at 1215 cm^−1^ is related to the PO_2_ asymmetric stretching vibration [71].

ATR-FTIR spectra of blank ALG NPs showed the similar features as the spectrum of sodium alginate with slight differences. Initially, the increasing intensity of the peak at 3334 cm^−1^ shows that the crosslinking has been employed effectively [74]. A slight shift from 1408 cm^−1^ to 1416 cm^−1^ was observed for the peak related to carboxyl asymmetric stretching. This difference results from the complexation of alginate carboxyl and the calcium ions to form calcium alginate gel [75].

Several changes can be noted following loading CRISPR pDNA onto ALG NPs, indicating that interactions between pDNA and ALG NPs occurred. The changes in band positions can also be identified in the nucleobase region. The band at 1614 cm^−1^ in the CRISPR ALG NPs spectrum, which is probably related to the combination of bands at 1644 cm^−1^ and 1555 cm^−1^ from DNA, and 1594 cm^−1^ from calcium alginate, indicating the ALG probably interacts with the CRISPR pDNA through hydrogen bond forming to the nucleobases. Tween 20 band at 1734 cm^−1^ appears in blank ALG NPs and CRISPR ALG NPs, indicating that Tween 20 remains in the formula and did not completely leak into collection solution.

Moreover, CRISPR pDNA and blank ALG NPs spectra were combined in silico using the SpectraGryph 1.2 spectroscopy software. The resultant merged spectrum was used to confirm the changes in spectrum following loading CRISPR pDNA onto ALG NPs. In silico CRISPR ALG NPs showed a new band at 1582 cm^−1^, which is probably related to the combination of bands at 1644 cm^−1^ and 1555 cm^−1^ from DNA, and 1594 cm^−1^ from calcium alginate.

Thus, FTIR result reveals that pDNA, ALG, Tween 20, and calcium chloride all are involved in nanoparticle formation with ionic complexation between calcium and alginate carboxyl groups, in addition to physical interaction between pDNA and calcium alginate through hydrogen bonding.

#### 2.2.4. Encapsulation Efficiency

Encapsulating hydrophilic substances in alginate nanoparticles can be challenging due to the fact that the encapsulated material can easily leak out to the gelling bath. The hydrogel structure of alginate nanoparticles usually has large pores that facilitate the movement of water-soluble small molecules out from the particles. For example, EE of paracetamol in alginate beads dropped rapidly from 33.92 ± 1.12% at 10 min gelation time to as low as 5 ± 1.32% at 30 min [76]. Similar findings were reported for macromolecules, such as whey protein, in which the EE achieved was below 60% [47]. However, the optimized NP formulation in the current study successfully encapsulated almost all the pDNA (EE = 99.94 ± 0.10%). This can be explained by the strong crosslinking of the external shell of the NPs. The crosslinking is crucial in context of encapsulation using alginate polymers. Chan and colleagues supported this phenomenon by assaying various types of alginates with different mannuronic/guluronic acid ratios, achieving higher EE values for alginates with higher guluronic acid content [77]. Besides, one advantage of electrospray method is producing particles with high EE. The high EE was reported in literature on electrospraying, where it was found to be more than 80%. Wang and team reported 82% bovine serum albumin protein EE in PLGA microparticles [78], while Xu and colleagues encapsulated bovine serum albumin in polylactide particles with 80% efficiency [64].

#### 2.2.5. pDNA Integrity After Encapsulation

The results revealed that 5X tris-acetate-edetate (TAE) buffer successfully extracted the pDNA from CRISPR ALG NPs. The extracted pDNA migrated similar to the control, un-encapsulated CRISPR pDNA, as shown in Figure 6. Additionally, no change in supercoil conformation was observed after extraction of pDNA, indicating that ALG NPs protected the pDNA and preserved its integrity. This demonstrates that the electrospray technique is relatively gentle process. Hence, it has been used to fabricate plasmid [63], protein [62], and enzyme [79] nanoparticles.

#### 2.2.6. pDNA Serum Stability

Naked pDNA degraded in 10% *v/v* FBS medium in the first hour, revealing clear smearing in agarose gel (Figure 6). On the other hand, ALG NPs partially protected the pDNA from degradation in the first hour, as little smearing was observed. However, after 24 h, ALG NPs were unable to protect the encapsulated pDNA, as smearing was observed. This can be attributed to the release of pDNA from the nanoparticles. Swelling of the particles in the medium could also facilitate the degradation of pDNA, where the pores in the gel structure become larger, permitting exchange of materials between the particles and the medium.

#### 2.2.7. In Vitro Release Profile

The in vitro release data of CRISPR pDNA from ALG NPs over 8 days’ time period is shown in Figure 7. CRISPR ALG NPs showed initial burst release, as 3.56% and 33.6% of pDNA released in 3 h and 24 h, respectively. However, the release rate was shown to prolong after that, wherein about cumulative 40% of the total loaded pDNA was released at 8 days. These results clearly indicate that CRISPR pDNA displays prolonged release, which is required in gene delivery to maintain the therapeutic dose. Figure 6 outlines the stability of CRISPR pDNA on agarose during the release study. The pDNA released at the 12 and 24 h time point showed one band closed to the position of the control pDNA. However, the amount released at the 8 days’ time point was under the detection limit of AGE.

According to Azad et al. (2020), the drug release from alginate beads relies on the dissolution medium penetration into alginate beads, swelling, and dissolution of the alginate hydrogel, as well as the dissolution of the encapsulated drug subsequent to leakage through the swollen hydrogel [80]. In this regard, the mechanism of swelling and dissolution of the alginate hydrogel involves release by the diffusion (from the hydrogel swollen pores) and erosion (from the dissolved position) due to calcium depletion from the hydrogel. Alginate hydrogels can be dissolved by the release of calcium ions through exchange reactions with sodium ion in the release medium. This converts insoluble calcium alginate to a soluble salt of sodium alginate, and the matrix will swell and disintegrate, resulting in the release of DNA. However, as the alginate salt used is high guluronic acid (high G), the release was slow. Guluronic acid conformation gives a high degree of coordination of the calcium, thereby forming rigid gels that are less prone to swelling and erosion.

#### 2.2.8. NP Cytotoxicity

The purpose of the cytotoxicity experiment was to demonstrate the safety of the CRISPR loaded NPs on HepG2 cell line over range of concentrations (1000, 500, 250, 125 µg/mL). The toxicity of CRISPR plasmid, blank ALG NPs, and CRISPR ALG NPs was evaluated by means of cell metabolic activity corresponding to the cell viability. Interestingly, cells treated with alginate encapsulating CRISPR plasmids yielded outstanding cell metabolic activity, of more than 100%, indicating that the NPs was not cytotoxic; instead, they increased the cells metabolic activity and supported cell growth over the studied concentrations range (Figure 8), even though unloaded alginate NPs and CRISPR plasmids induced toxicity into HepG2 cells. This might be attributed to the interaction between pDNA and alginate when they are combined to make CRISPR ALG NPs, which probably decreased the cytotoxicity of both of them. Additionally, the cells seemed to proliferate more quickly when they were treated with higher CRISPR ALG NP concentration, even though a larger amount of pDNA was applied to the cells. This might be due to the higher uptaken particles, which probably influence the mitochondrial activity of the cells [81]. This agreed with a previous report, which proved that HEK293 cells proliferated more quickly when they were treated with plasmid loaded alginate NPs [82]. Moreover, the majority of the studies that investigated potential toxicity issues associated with alginate NPs as gene carrier report safe use. For instance, Saeed and colleagues reported no significant difference in HepG2 cell viability after it was treated with Blank ALG NPs at concentration range from 1 to 30 µg/ mL [83]. Amiji and co-worker also reported no cytotoxic effect neither for alginate microspheres nor EGFP plasmid loaded alginate microspheres in J774a.1 murine macrophage at concentration 1000 µg /mL [84]. Moreover, there was significant difference in metabolic activity of cells treated with CRISPR ALG NPs and cells treated with pDNA: reagent (pDNA complexed with NanoJuice transfection reagent at ratio 1 µg:2.5 µL), whereby cells metabolic activity was less than 60% at all studied concentrations. The result, therefore, revealed CRISPR ALG NP is a safe system for delivery gene-editing plasmids in HepG2 cells at concentration range (125–1000 µg /mL).

#### 2.2.9. Cellular Uptake

Contrary to chemical drug delivery, in which the therapeutic agent can still be reached after drug release from carrier even if the carrier fails to be internalized into the cell, gene therapy can be executed only if the nucleic acid molecule can be brought into the target cell. Thus, the first barrier for gene delivery carrier is cell internalization. In order to assess whether the CRISPR ALG NPs were actually internalized efficiently, HepG2 cells were incubated with various concentrations of CRISPR ALG NPs added with coumarin-6 (NP properties are comparable to NPs without coumarin-6, as shown in Appendix A).

Coumarin-6-loaded NPs targeted the HepG2 cells in a concentration- and time-dependent manner. The intracellular fluorescence of coumarin-6-loaded NPs continued to increase with extended incubation time, indicating increased the up-taken quantity into the cell (Figure 9B). Similarly, the cellular uptake efficiency increased with extended incubation time (Figure 9C). The cellular uptake efficiency increased slightly, but significantly at all studied concentrations except 250 µg/mL, when incubation time changed from 30 min to 1 h. Nevertheless, a sharper and significant change was detected when incubation time was prolonged to 2 h. As time prolongated to 2 h, CRISPR ALG NPs uptake efficiency was increased by 1.51-, 2.95-, 2.18-, and 8.40-fold for concentrations 125, 250, 500, and 1000 µg/mL, respectively. Hence, the concentration affects the rate of increase. Higher uptaken quantity and efficiency have been achieved with higher concentration. This may indicate compatibility of alginate polymer with HepG2 cells. Previous study with gold showed saturation of HepG2 cells uptake [85]. In some cases, there was also exocytosis, especially when using polymer, like PLGA. For example, 65% of the internalized PLGA fraction to smooth muscle cells were undergoing exocytosis in 30 min [86].

Regardless of the endocytosis pathway of the CRISPR ALG NPs, the rate and extend of cellular uptake is mainly governed by NPs’ physiochemical characteristics (size, shape, and charge) [87]. Thus, higher cellular uptake efficiency (~80%) could be achieved at higher concentration (1000 µg/mL) at 37 °C for 2 h (Figure 9A). Qualitative analysis showed also that, at 100 µg/mL concentration, almost all cells had taken up ALG NPs.

#### 2.2.10. In Vitro Transfection Efficiency

Endocytosis pathway is the main obstacle during carrier internalization because most of the DNA is retained in the endosomes/lysosomes and is eventually degraded or inactivated. Thus, the efficient carrier must destabilize or escape the endosome, consequently allowing the pDNA to escape into the cytoplasm before lysosomal degradation can take place. Therefore, HepG2 cells were transfected with two CRISPR plasmids (pSpCas9(BB)-2A-miRFP670 containing the reporter gene, RFP, and gRNA_GFP-T2) and analyzed after 48 and 72 h. CRISPR ALG NPs transfection efficiency was compared with NanoJuice transfection reagents (a dendrimer and polycation liposome-based reagent).

The red fluorescence from the transgene was observed in the transfected cells in all treated conditions (Figure 10A). The rate of CRISPR ALG NP transfection was concentration- and time- dependant. Higher efficiency was achieved with higher concentration, 9.09 ± 4.05% and 25.06 ± 0.98% for 100 and 1000 μg/mL NP concentration, respectively (*p* < 0.05). This increase in transfection efficiency could be due to the increase in the cellular uptake. The cellular uptake of CRISPR ALG NPs was 2.18-fold more at concentration 1000 μg/mL, compared to 100 μg/mL. Moreover, the transfection efficiency increased significantly by 5.6-fold, from 9.09 ± 4.05% to 50.88 ± 8.67%, as the transfection time was prolonged from 48 to 72 h (*p* < 0.05) (Figure 10B). This could be related to endosomal swelling, elicited by CRISPR ALG NPs, which reaches its peak after 48 h incubation, followed by the sustained release of encapsulation pDNA into cytosol (Figure 11). It is hypothesized that the endosomal escape of the plasmid loaded calcium alginate nanoparticles happens between 12–48 h after transfection [88]. Even though the exact mechanism of CRISPR ALG NPs in escaping endosome/lysosome is not well studied here, it was suggested that the crosslinking calcium ion from calcium alginate particles may be sequestered by intracellular citrate or phosphate ions; hence, alginate and its hydrogel’s dissolution product exhibits a ‘proton sponge effect’ that increases the osmotic pressure and facilitates the rupture of endosome/lysosome, thereby increasing the endosomal release of pDNA into the cytosol [89].

In comparison with other non-viral transfection reagent, alginate-based gene delivery shows better transfection efficiency at 72 h. As a positive control for transfection, NanoJuice reagent was used at the commonly used ratio of 1 μg pDNA to 2.5 μL (2 μL transfection core and 0.5 μL transfection booster). NanoJuice reagent showed 14.25 ± 0.66% and 33.25 ± 2.15% efficiency, at 48 and 72 h post-transfection, respectively (*p* < 0.05). This demonstrated that CRISPR ALG NPs exhibited better delivery ability for large-size plasmid than cationic lipid transfection reagent with prolonged action, owing to its capability to encapsulate pDNA completely, hence shielding it from intracellular environment and release it in prolonged manner.

Further, RFP expression 48 h post-transfection by CRISPR ALG NPs was about four-fold lower than NanoJuice reagent (*p* < 0.05); however, the expression after 72 h was approximately same (Figure 10C), suggesting the successful delivery of ALG NPs as a carrier for CRISPR plasmids.

Our lipid transfection results are slightly lower than that reported in the literature for HepG2 cells at relatively the same incubation time (~2 days) and number of transfected cells, plasmid amount, and transfection reagent to plasmid ratio. For instance, transfection of HepG2 with pcDNA3-EGFP plasmid (with size ~6000 bp) using lipofectamine LTX (3 µg TR to 1 µg pDNA ratio) is reported to result in about 15% transfection efficiency. This could be attributed to the size of plasmid, as the it is one of the main influencing factors during transient transfection assay [90,91]. Xu and colleagues proved that the same carrier that shown efficient EGFP plasmid has also shown low efficient for delivery of a relatively larger plasmid (8–10 kb) against chondrocytes. Thus, herein, the reduced efficiency achieved could be related to delivering two plasmid of size 9175 and 3973 bp.

#### 2.2.11. GFP Gene Disruption

The targeted gene editing activity of CRISPR plasmids through non-homologous end joining (NHEJ) was evaluated by Genome Cleavage Detection Assay (T7EI cleavage assay) to detect the indels. HepG2 were transfected with either Cas9 or with Cas9, along with gRNA. Transfection with the Cas9 plasmid alone did not mediate cleavage of the target sequence in the cells, as the PCR amplicon size was similar to that of purified EGFP plasmid at around 850 bp. However, Cas9 mediated gene editing when Cas9 plasmid was delivered, along with the gRNA plasmid, in ALG NPs, allowing cleavage of the target sequence in GFP plasmid. After the cleavage the cell tends to repair the cleaved DNA through NHEJ, which can be detected trough the cleavage detection assay, which detects the mismatch in the PCR fragments. Once mismatch is detected, the enzyme will cut the PCR piece. As shown in Figure 12, the assay resulted in two fragments of approximately 500 bp and 350 bp. Moreover, the efficiency of indel formation was calculated by comparing the intensities of cleaved and uncleaved fragments using ImageJ software. The indel formation efficiency of ALG NPs-mediated delivery was 3.37%, indicating that CRISPR plasmids delivered to the cells via ALG NPs still active and can induce Double strand break (DSB) in target DNA. Nevertheless, the gene editing efficiency is relatively low and needs to be improved before using this delivery vehicle for therapeutic purpose. The improvement might be done through coating of ALG NPs with cationic polymer. Compared to the other polymers, chitosan has been appealing as surface coating for a carrier. Several in vitro and in vivo studies have concluded that presence of these polymers as ligand on nanocarrier surface can improve physicochemical stability exposure to different conditions, promote mucoadhesiveness and tissue penetration, and modulate cell interactions [92].

Moreover, to confirm that CRISPR/Cas9 mediated successful gene editing, the cells were imaged, and the mean fluorescence intensity of GFP delivered to HepG2 either expressing CRISPR system or not was measured. As CRISPR system is targeted against GFP-T2 sequence, once the cells are transfected with GFP plasmid, CRISPR ribonucleoprotein RNP attacks GFP plasmid, binds to its target, and indels the plasmid, resulting in unfunctional protein (protein will lose the green fluorescence) [93]. The mean fluorescence intensity of GFP was reduced by 0.49-fold after 2 days of delivering GFP plasmid to the cells, compared to negative control cells (cells did not express CRISPR system). This could indicate the activity of Cas9/gRNA in gene editing of the target gene.

## 3. Materials and Methods

### 3.1. Material

High stiffness gelation sodium alginate (grade IL-6G, low molecular weight, high G, 30-60 mPa.S) was obtained from Kimica (Tokyo, Japan). Calcium chloride and Tween 20 were obtained from Merck (Darmstadt, Germany). GFP plasmid (pcDNA3-EGFP) was a gift from Doug Golenbock (Addgene plasmid #13031), pSpCas9(BB)-2A-miRFP670 encoding Cas9 tagged with red fluorescence protein (RFP) was a gift from Ralf Kuehn (Addgene plasmid #91854) and gRNA_GFP-T2 encoding gRNA that targets specific location in the green fluorescence protein (GFP) gene was a gift from George Church (Addgene plasmid #41820) [94]. Low glucose Dulbecco’s Modified Eagle Medium (DMEM) and phenol red free DMEM were obtained from Nacalai Tesque, Inc. (Kyoto, Japan). Foetal bovine serum (FBS) was purchased from Tico Europe (Amstelveen, Netherlands). 3-(4,5-Dimethylthiazol-2-yl)-2,5 diphenyltetrazolium bromide (MTT) was purchased from Life Technologies (Grand Island, NY).. NanoJuice Core Transfection Reagent was from Sigma-Aldrich (St. Louis, MO, USA). Hoechst 33342 live cell stain was from GeneCopoeia (Rockville, MD, USA).

### 3.2. Encapsulation of CRISPR pDNA in ALG NPs

The two CRISPR plasmids were propagated from their agar stabs upon receiving, pSpCas9(BB)-2A-miRFP670 encoding Cas9 was propagated in *Escherichia coli* DH5α, while gRNA_GFP-T2 encoding gRNA was in *Escherichia coli* Top10, and purified using a PureYelid maxiprep plasmid kit (Promega, Madison, WI, USA) according to the manufacturer’s instructions. Each plasmid concentration was adjusted to 300 µg/mL in sterile nuclease free water. Then solution of the two CRISPR plasmids were prepared by mixing equal volumes to obtain the final CRISPR plasmids solution of 300 µg/mL (150 µg each).

Different concentrations of sodium alginate solution were freshly prepared in sterile deionized water and characterized for physiochemical properties using a conductometer (FiveEasy FE30, Metler Toledo, Switzerland), rheometer (Mars, HAAKE, Germany), and surface tension analyser (SEO, Sigma 703D). Table 1 shows the content of each solution.

CRISPR plasmids solution and selected alginate solution (S8) were mixed at variable pDNA:ALG weight ratio, 1:100, 10:100, and 20:100. CRISPR pDNA ALG solutions were loaded in a 1 mL plastic syringe connected to a 30G stainless-steel needle through a silicon tube. The syringe was placed horizontally on a syringe pump (Shenchen SPLab02, Baoding, China) to control the flow rate. The positive electrode of a high voltage power supply (Analog Technologies, Inc., San Jose, CA, USA) was linked to the vertical needle tip. The collector was a fixed glass plate covered with a grounded aluminum foil and containing 10 mL 1.5% *w/v* calcium chloride solution (pH = 3.75 ± 0.02). The spraying distance between the needle tip and gelation bath surface was fixed at 4 cm (Figure 1). The NPs were left for curing in the gelling bath for 15 min with stirring, and then they were collected by centrifugation at 14,000 rpm for 30 min. Next, they were washed by distilled water and resuspended in distilled water. All of the electrospray experiments were carried out at room temperature under ambient air conditions.

To select the best ratio, CRISPR pDNA ALG solutions were electrosprayed at fixed flow rate of 0.1 mL/h and operation voltage of 9.5 kV. The NPs were left for curing in the gelling bath for 15 min with stirring, and then the encapsulation efficiencies of resultant NPs were determined by indirect method via quantification of pDNA portion that remained in gelling medium using Quantus™ Fluorometer (Promega, Madison, WI, USA). The best ratio was selected based on the highest encapsulation efficiency.

### 3.3. Optimization of CRISPR ALG NPs

Two factors three levels (3^2^) full factorial design was used to study the effect of process variables, namely voltage (9.5, 11, and 12.5 kV) and flow rate (0.1, 0.3, and 0.5 mL/h) on the median diameter (Di50) of the nanoparticles. The distance between nozzle tip and gelling bath was fixed at 4 cm. In addition, the formulation variables were also fixed as alginate concentration: 1% *w/v*, Tween 20 concentration: 1% *v/v* and pDNA loading to alginate weight ratio: 10 to 100. Results from this design were used as foundation to predict the desirable response (Di50). Priority for optimization were in the form of minimum particle size.

### 3.4. Characterzation of Optimized CRISPR ALG NPs

#### 3.4.1. Particle Size and Zeta Potential Analysis

Particle size and zeta potential were determined by dynamic light scattering (Zetasizer Nano-S and Nano-Z, Malvern Instrument, Worcestershire, UK). Analysis was performed at a scattering angle of 90° for the duration of 10 s, 12–16 times, at 25 °C. Blank ALG NPs and CRISPR ALG NPs were suspended in distilled water or 10% *v/v* FBS at concentration of 5 µg/ mL. Blank ALG NPs were prepared exactly like the CRISPR ALG NPs but without including plasmid solution in the formulation.

#### 3.4.2. TEM

The nanostructure (size and surface morphology) of NPs was observed by Libra 120 transmission electron microscope (Zeiss, Oberkochen, Germany). The sample was prepared by dropping the NP suspension from the gelling bath onto a carbon coated copper grid, without staining or fixation, and then drying it in an incubator at 37 °C.

#### 3.4.3. ATR-FTIR

ATR-FTIR spectra of lyophilized blank ALG NPs and CRISPR ALG NPs were obtained by scanning from 400 to 4000 cm^−1^ at 4 cm^−1^ resolution at room temperature using Perkin Elmer 100 spectrophotometer (Perkin Elmer Corp., Norwalk, CT, USA). Blank ALG NPs were prepared exactly like the CRISPR ALG NPs but without including plasmid solution in the formulation. Blank ALG NPs and CRISPR ALG NPs were recovered from gelling bath and washed by distilled water to remove the excess amount of calcium chloride by centrifugation at 14,000 rpm for 30 min. Then, CRISPR ALG NPs were lyophilized using a Christ Freeze Dryer (Martin Crist Alpha 1–2LD Plus, Pocklington, UK) without cryoprotectant. The primary drying was carried out at −55 °C for 12 h, during which the pressure was reduced to 10^−2^ mbar. The sample was then attached to a secondary drying manifold, dried overnight, and then sealed under vacuum.

#### 3.4.4. Encapsulation Efficiency

The encapsulation efficiency was calculated by the indirect method based on the quantification of un-encapsulated pDNA. After NP fabrication, the gelling medium containing NPs was collected, and the supernatant was obtained by centrifugation at 14,000 rpm for 30 min. The pDNA portion present in the gelling medium was measured using Quantus™ Fluorometer (Promega, Madison, WI, USA) following the manufacturer protocol. The amount of encapsulated pDNA was calculated by using Equation (1):(1)$$\begin{array}{c} \mathrm{Percentage}\;\mathrm{of}\;\mathrm{encapsulated}\;\mathrm{pDNA}=\\ \frac{\mathrm{total}\;\mathrm{amount}\;\mathrm{of}\;\mathrm{pDNA}\;\mathrm{in}\;\mathrm{the}\;\mathrm{formulation}-\mathrm{free}\;\mathrm{amount}\;\mathrm{of}\;\mathrm{pDNA}\;\mathrm{in}\;\mathrm{gelling}\;\mathrm{medium}\;}{\mathrm{total}\;\mathrm{amount}\;\mathrm{of}\;\mathrm{pDNA}\;\mathrm{in}\;\mathrm{the}\;\mathrm{formulation}\;}\\ \times 100 \end{array}$$

#### 3.4.5. pDNA Integrity After Encapsulation

To evaluate the process of pDNA encapsulation, the integrity of pDNA was checked by agarose gel electrophoresis (AGE). The encapsulated pDNA was extracted out from NPs by suspending an amount of NPs containing 1 µg pDNA in 100 µL of 5× TAE buffer. The extracted pDNA was loaded onto 1% *w/v* agarose gel and then run for 60 min at 120 V.

#### 3.4.6. pDNA Serum Stability

The CRISPR ALG NPs were examined for the ability to protect pDNA from serum by AGE. CRISPR ALG NPs was suspended in 10% *v/v* FBS in concentration of 100 µg/mL and the mixture was incubated for 1 h or 24 h at 37 °C. The positive control was naked pDNA, and it was treated as the CRISPR ALG NPs. The sample and the control were loaded on 1% *w/v* agarose and run for 60 min at 120 V [95].

#### 3.4.7. In Vitro Release Profile

The release profile study was carried out by resuspending the CRISPR ALG NPs in phosphate buffered saline (PBS) pH 7.4. A quantity of NPs containing 10 µg pDNA was collected from gelling bath and washed by distilled water by centrifugation at 14,000 rpm for 30 min. CRISPR ALG NPs were resuspended in 5 mL PBS and incubated at 37 °C. At predetermined time points, the NPs were centrifugated at 14,000 rpm for 30 min, and the supernatant was collected and then subjected to fluorometer to quantify the released pDNA. The collected NPs after centrifugation were resuspended and replaced with fresh PBS. Additionally, the integrity of released plasmid was checked by AGE using 1% *w/v* agarose gel run for 60 min at 120 V [95].

#### 3.4.8. Cell Culture

Hepatocellular carcinoma (HepG2) cell line (ATCC^®^ HB-8065™) was purchased from American Type Culture Collection (Manassas, VA, USA). HepG2 cells were cultured according to the supplier protocol. The cells were maintained in DMEM, supplemented with 10% *v/v* FBS, 100 IU/mL penicillin, and 100 µg/mL streptomycin. The cells were grown in T-25 flasks (CellStar, Greiner bio-one, Kremsmunster, Austria) and incubated at 37 °C, 5% CO_2_.

#### 3.4.9. Nanoparticle Cytotoxicity

Cell viability was determined to assess the in vitro cytotoxicity of CRISPR ALG NPs to HepG2 cells by determining the metabolic activity of treated HepG2 cells compared to control cells via MTT assay. Briefly, HepG2 cells were seeded into 96-well transparent flat-bottom plate at a density of 1 × 10^4^ cells/well. Cells were incubated in 100 μL of complete growth media at 37 °C and 5% CO_2_. After 24 h, the media was removed and cells were incubated in 100 μL of media suspended with nanoparticles at different concentrations (1000, 500, 250, 125 µg/mL). After 48 h of NP exposure, media was replaced with 90 µL phenol red-free complete media and 10 µL of MTT reagent (5 mg/mL). After 4 h, the medium was removed, and 100 μL of DMSO was added in each well. The plate was incubated at room temperature (until crystals were dissolved), and the absorbance was recorded at 570 nm using a Tecan Infinite 200 microplate reader (Tecan Austria GmbH, Grodig, Austria). Percent relative cell metabolic activity values were calculated, after blank subtraction, by dividing the absorbance of treated cells by the untreated ones. The negative control was untreated HepG2 cells, while pDNA complexed NanoJuice transfection reagent (1 μg: 2.5 μL) was used as the positive control.

#### 3.4.10. Cellular Uptake

The interaction between ALG NPs and the cells was evaluated by studying the cellular uptake. The concentration- and time-dependent cellular uptake was determined in HepG2 cell line using coumarin-6 as a fluorescent probe to label ALG NPs (2.2 µg coumarin-6 to 100 µg NPs). Since coumarin does not dissolve in water, 2.2 µg coumarin-6 was dissolved in 0.2 uL methanol and then added to 0.1 uL corn oil. The oil was then emulsified in 100 µL optimized alginate solution. Homogenization of emulsion was done by using ultra sonic processor (QSonica, 53 Church Hill Rd. Newtown, PA, USA; 6 cycles for 15 s each at 20 MHz). Next, CRISPR pDNA was added with weight ratio 1:10 (pDNA: ALG). Lastly, NPs were prepared using the electrospray method mentioned above. The particles were characterized for their size, charge, and morphology and compared to unlabeled counterparts. HepG2 cells were seeded into 96-well transparent flat-bottom plate at a density of 1 × 10^4^ cells/well. Cells were incubated in 100 μL of complete growth media for 24 h at 37 °C and 5% CO_2_. Then cells were incubated with a suspension of coumarin-6 labeled NPs at different concentrations (1000, 500, 250, and 125 µg/mL) for different time periods (30, 60, and 120 min). After incubation, the cells were washed three times with ice-cold PBS to remove the noninternalized nanoparticles [96]. The cells were subsequently trypsinized and dissolved in 100 µL of Promega lysis reagent. Finally, the fluorescence intensity was measured by fluorometry (λex = 490 nm, λem = 505 nm). In order to calculate the cellular uptake efficiency, a series of NPs were used to prepare the calibration curves (fluorescence intensity versus concentration) in the same lysis medium. Moreover, in a parallel experiment, the cellular uptake images were recorded at 2 h incubation using the Cytell cell imaging system (GE Healthcare Life Science, Buckinghamshire, UK) after staining the nuclei with Hoechst 33342 live cell stain.

#### 3.4.11. In vitro Transfection

CRISPR plasmids [94] (encoding Cas9-RFP and the gRNA against GFP)-loaded alginate nanoparticles were used to transfect the cultured HepG2 cells. Briefly, cells were seeded in 96-well transparent flat-bottom plate at a density of 1 × 10^4^ cells/well and grown overnight to reach a confluence of 70–80% before transfection. Then, media was replaced with 100 μL of media suspended with nanoparticles at concentration of 100 µg/mL and incubated at 37 °C and 5% CO_2_. After 2 h of exposure to the nanoparticles, media was replaced with 100 µL complete media and incubated for 48 h. Moreover, the cells were transfected with NanoJuice as a positive control. The transfection mixture was prepared by mixing 1 μg pDNA to 2 μL transfection core and 0.5 μL transfection booster according to manufacturer instruction.

#### 3.4.12. Quantification of Transfection Efficiency

At 78 or 72 h post-transfection, the nuclei of cultured cells were blue-stained by Hoechst 33342 live cell stain and viewed under Cytell cell imaging system (GE Healthcare Life Science, Buckinghamshire, UK) to detect the expression of Cas9-RFP. Successful transfection yields red fluorescence color. Blue and red channels photographs of random fields of HepG2 cells were taken at high magnification (field of view 880 × 660 µm). All imaging parameters, like exposure time and gain, were fixed for all photographs to obtain reliable images to be used for quantification. Three photographs for each channel were analyzed by two independent observers using ImageJ software (NIH, Bethesda, MD, USA). The transfection efficiency was calculated by dividing the number of RFP-positive cells by the total number of surviving cells in the photographs. [97]. In addition, the kinetic of gene expression was monitored for 48 and 72 h, and it was evaluated by quantifying the mean red florescence intensity using ImageJ software [95].

#### 3.4.13. GFP Gene Disruption

The targeted gene editing activity of two CRISPR plasmids delivered via ALG NPs to cleavage the GFP target gene sequence was detected by T7 endonuclease I (T7EI) assay using Alt-R™ Genome Editing Detection Kit (Integrated DNA Technologies, Coralville, Lowa, USA) according to the manufacturer protocol. The CRISPR ALG NPs (encapsulating both Cas9 and gRNA plasmids) were used to transfect the cultured HepG2 cells, as mentioned before. Cells were allowed for 24 h to express the ex-genes, which was confirmed with fluorescence microscope. Next, HepG2 cells were transfected with GFP plasmid using Nanojuice transfection reagent. The DNA from transfected HepG2 cells was extracted using InstaGene Matric (Bio-Rad, Hercules, California, USA), and the target T2 sequence (GGAGCGCACCATCTTCTTCA) was amplified using polymerase chain reaction (PCR) master mix, EGFP forward primer (5′-CCCACTGCTTACTGGCTTATC-3′) and EGFP reverse primer (5′- CCATGTGATCGCGCTTCT -3′) (First Base, Selangor, Malaysia) at the following conditions: 95 °C for 10 min; then 40 cycles of (95 °C for 30 S; 55 °C for 30 S and 72 °C for 30 S); then a final step of 72 °C for 7 min. Next, 10 µL of the amplified PCR product was mixed with 2 µL T7EI Reaction Buffer (10×) and 6 µL nuclease-free water. The heteroduplex were formed using a thermocycler with the following protocol: 95 °C for 10 min; 95 °C to 85 °C at −2 °C/s; 85 °C to 25 °C at −0.3 °C/s; holding for 1 min at each step. The heteroduplex products were then digested with T7EI enzyme (1 U/μL) for 1 h at 37°C. The digested product was subjected to 1% *w/v* agarose gel run for 60 min at 120 V. The integrated intensities of insertions and deletions (indels) were estimated using ImageJ software (NIH, Bethesda, MD, USA), and the indel formation efficiency was calculated using Equation (2), where a is intensity of uncleaved PCR fragment, and b and c are intensities of the cleaved fragments.
(2)$$\mathrm{indel}\;\left(\%\right)=100\left(1-\sqrt{1-\frac{b+c}{a+b+c}}\right).$$

To further confirm the gene editing using cell imaging, the cells were transfected with 1 μg CRISPR plasmids encapsulated within alginate nanoparticles into 96-well plates (1 × 10^4^ cells/well). After 48 h, CRISPR expressing HepG2 cells were re-transfected with 1 μg GFP plasmid using Nano Juice transfection reagent. Then, the cells were imaged under Cytell cell imaging system (GE Healthcare Life Science, Buckinghamshire, UK), and the mean fluorescent intensity of GFP was analyzed by ImageJ. HepG2 cell transfected with 1 μg GFP plasmid were used as control.

#### 3.4.14. Statistical Analysis

Design of experiment was performed using full factorial design (Minitab software, version 19). Evaluation of the model was carried out using ANOVA. All data presented as mean ± SD (*n* = 3). *P* < 0.05 considered as a statistically significant difference.

## 4. Conclusions

In summary, this study opens up a new avenue for generation of biocompatible alginate nanoparticles, which are capable for high loading and prolonged release of pDNA in one-step preparation. Moreover, nanoparticles prepared by this technique do not need further steps of purification and removal of other ingredients. Electrosprayed alginate particles delivered large size plasmid efficiently into mammalian cells, while preserving the integrity of the payload the during encapsulation process. We address three critical points: First, alginate nanoparticles internalization, visualized by presence of coumarin-6 labeled NPs; second, the successful plasmid transcription, visualized by red fluorescence resulting from RFP reporter gene expression; and third, the decrease of green fluorescence following GFP plasmid indel as result of CRISPR ribonucleoprotein activity. Further investigation of these particles would facilitate the clinical research employing alginate nanoparticles for biomedical applications.

## Figures and Tables

**Figure 1 pharmaceuticals-13-00158-f001:**
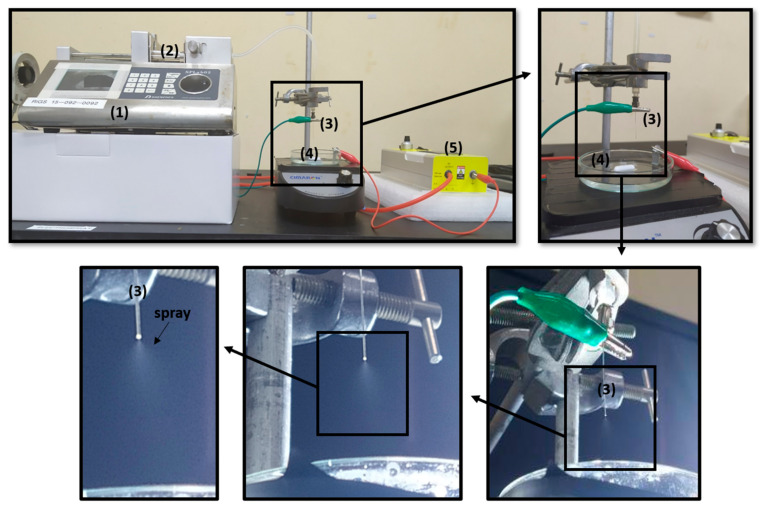
Experimental setup for electrospray fabrication of calcium alginate nanoparticles in cone jet mode: (**1**) syringe pump, (**2**) syringe, (**3**) nozzle tip, (**4**) collector (gelling bath), and (**5**) high voltage power supply.

**Figure 2 pharmaceuticals-13-00158-f002:**
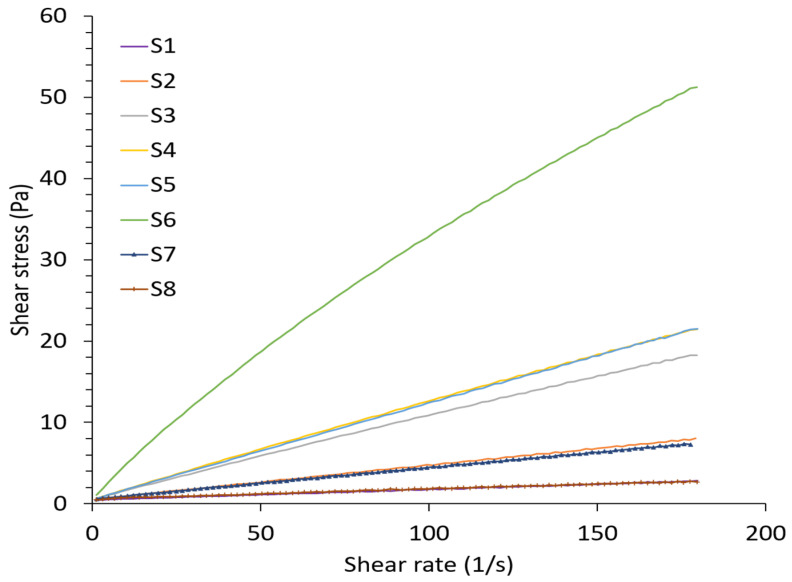
Scheme plot of shear stress versus shear rate of alginate solutions.

**Figure 3 pharmaceuticals-13-00158-f003:**
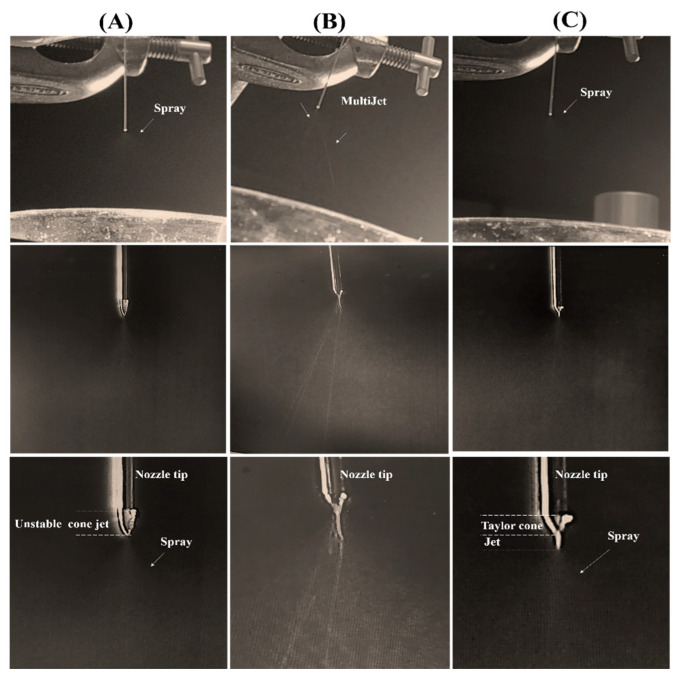
Camera image showing the spray mode by different ALG solutions: (**A**) unstable cone jet from S1, (**B**) unstable multijet from S3, and (**C**) stable cone jet from S8. Process parameters were nozzle tip: 30 G, flow rate: 0.1 mL/h, applied voltage: 9.5 kV, distance between tip and collector: 4 cm.

**Figure 4 pharmaceuticals-13-00158-f004:**
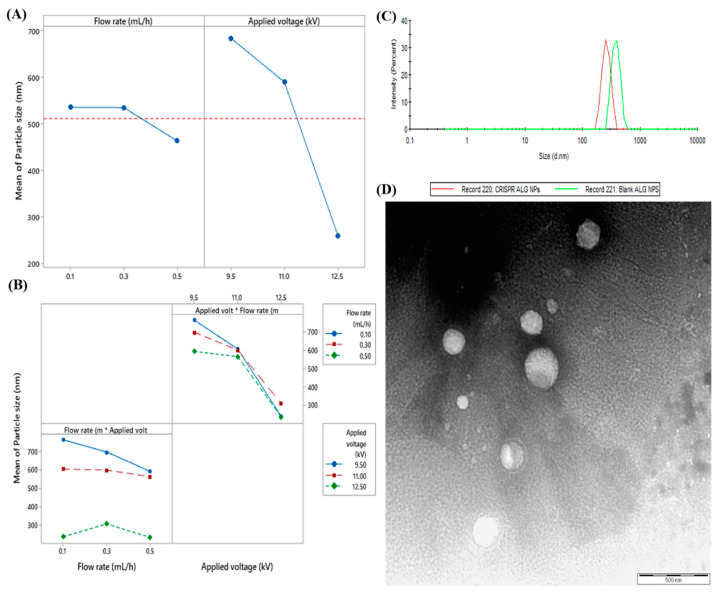
(**A**) The main effect plot of applied voltage and flow rate on the particle size. (**B**) The interaction effect of applied voltage and flow rate on the particle size. (**C**) Size distribution by intensity for optimized CRISPR ALG NPs and blank ALG NPs. (**D**) TEM image for CRISPR ALG NPs.

**Figure 5 pharmaceuticals-13-00158-f005:**
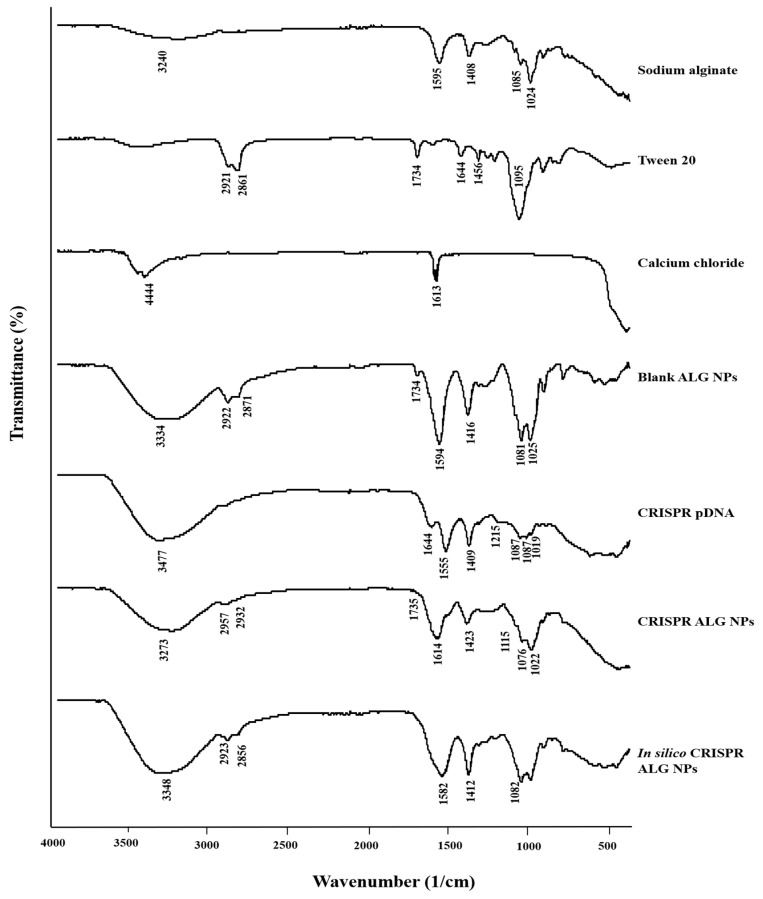
Attenuated Total Reflectance-Fourier Transform Infrared Spectroscopy spectra of sodium alginate, Tween 20, calcium chloride, CRISPR pDNA, blank ALG NPs, and CRISPR ALG NPs.

**Figure 6 pharmaceuticals-13-00158-f006:**
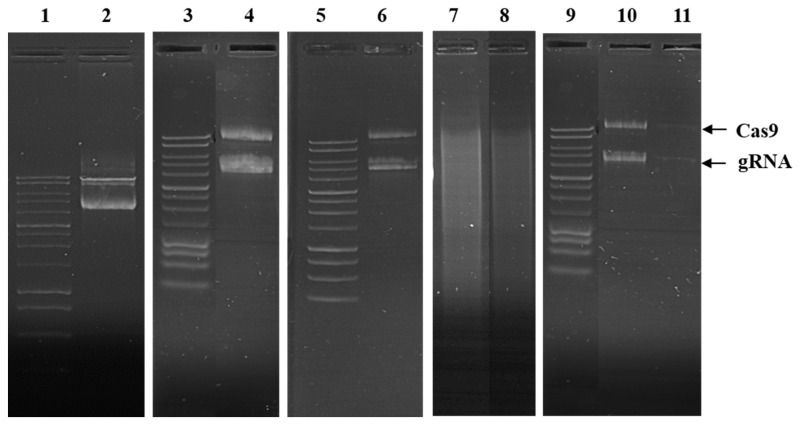
Integrity of CRISPR pDNA after encapsulation. Lanes: (**1**,**3**,**5**,**9**) are 1 kb ladder, (**2**) mixture of CRISPR plasmids (gRNA and Cas9), and (**4**) CRISPR plasmids (gRNA and Cas9) extracted from CRISPR ALG NPs. Stability of CRISPR ALG NPs in the presence of serum after 1 and 24 h incubation; (**6**,**8**) are CRISPR ALG NPs incubated in 10% *v/v* fetal bovine serum (FBS) for 1 and 24 h, respectively, (**7**) mixture of naked CRISPR pDNA, gRNA, and Cas9 incubated in 10% *v/v* FBS for 1 h. Stability of released CRISPR pDNA from ALG NPs; (**10**,**11**) are the released CRISPR pDNA at 12 and 24 h time point.

**Figure 7 pharmaceuticals-13-00158-f007:**
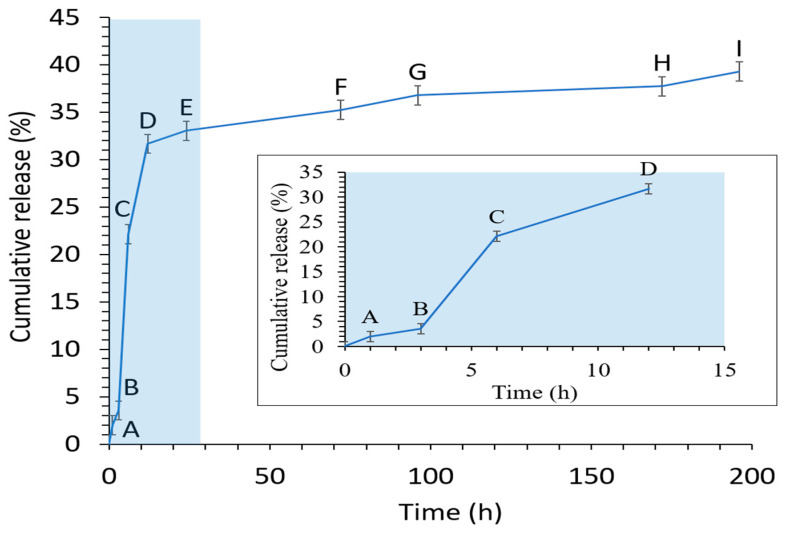
In vitro CRISPR pDNA cumulative release from CRISPR ALG NPs (mean ± SD; *n* = 3) in phosphate buffered saline (PBS) (pH 7.4). The inserted image represents detail of release during the first 12 h.

**Figure 8 pharmaceuticals-13-00158-f008:**
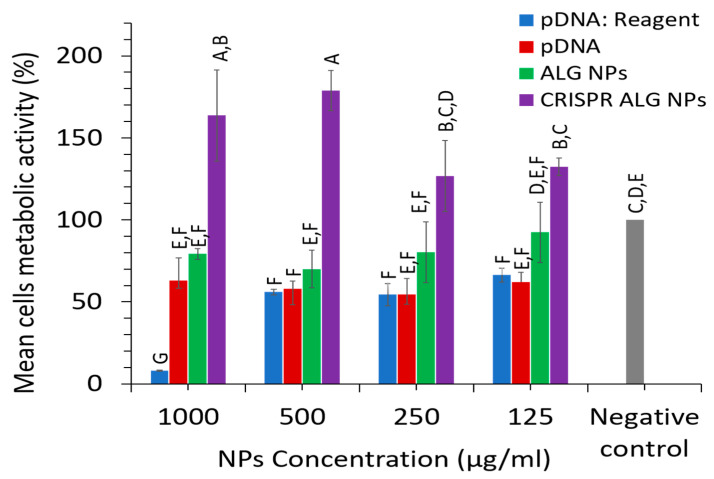
Viability of HepG2 cells 48 h post-transfection (mean ± SD; *n* = 3). pDNA: ALG ratio 1:10. Concentration of pDNA is equivalent to the concentration of pDNA in NPs. pDNA: NanoJuice transfection reagent ratio 1:2.5. Statistical significance of results was determined using Tukey’s Multiple Comparison Test. Letter denotes significance, whereby groups that do not share a letter are significantly different *p* < 0.05 (*n* = 3).

**Figure 9 pharmaceuticals-13-00158-f009:**
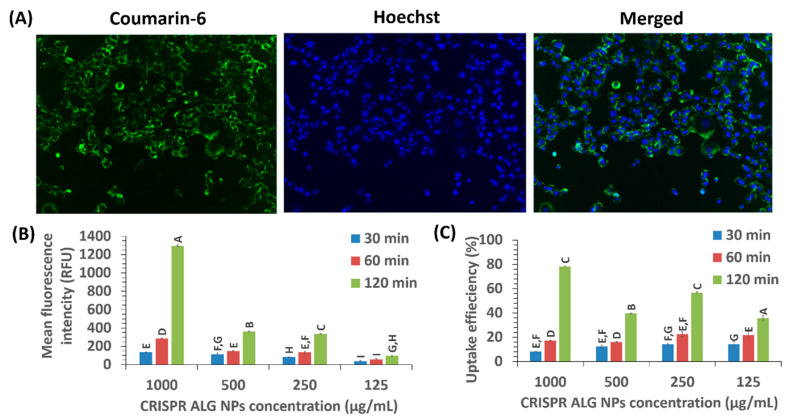
(**A**) The cellular uptake of coumarin-6 loaded NPs in HepG2 cells at concentration 100 µg/mL at 37 °C for 2 h. The cell nuclei stained with Hoechst, and green color was the intrinsic fluorescence from courmarin-6. (**B**) The mean fluorescent intensity of NPs by HepG2 cells at various concentrations for 30, 60, 120 min. (**C**) Cellular uptake efficiency of NPs by HepG2 cells at various concentrations for 30, 60, 120 min. Letter denotes significance, whereby groups that do not share a letter are significantly different *p* < 0.05 (*n* = 3).

**Figure 10 pharmaceuticals-13-00158-f010:**
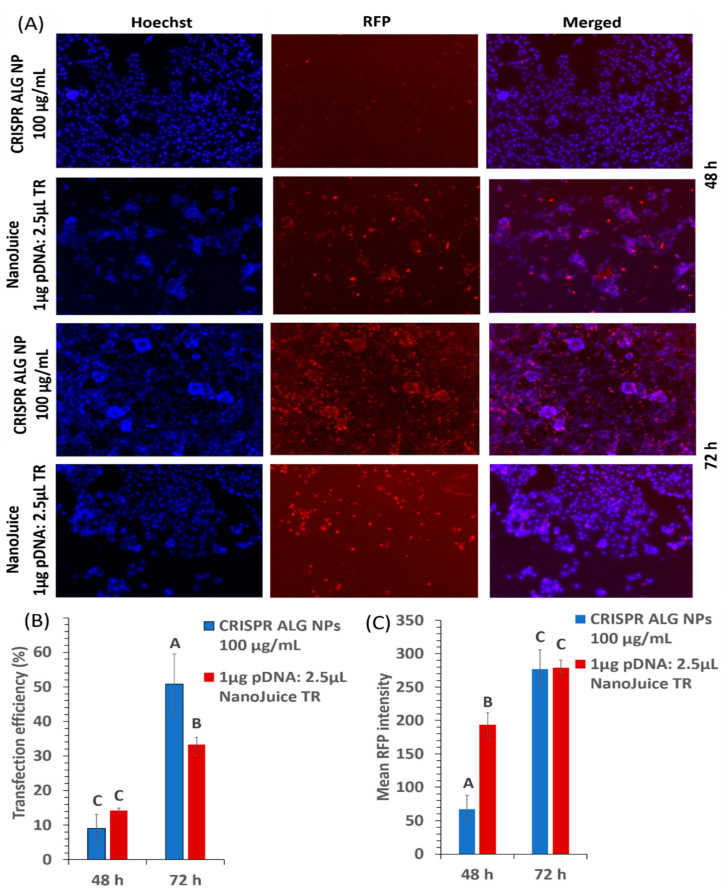
(**A**) Red fluorescence protein (RFP) fluorescence in HepG2 cells 48 h after transfection of RFP-expressing CRISPR plasmid. Panels show blue and red channel for CRISPR ALG NPs, NanoJuice reagent. All photographs were taken at high magnification (field of view 880 × 660 µm). (**B**) Transfection efficiencies by different reagents in HepG2 cells. Bars represent the means ± SD of 3 random cell field photographs at 48 and 72 h. CRISPR ALG NPs was containing CRISPR plasmids in weight ratio 1:10 (pDNA:ALG). The amounts of DNA plasmids used was 1 μg were combined with 2 μL reagent NanoJuice core reagent and 0.5 μL NanoJuice transfection booster. (**C**) Mean red fluorescence intensity expression by different reagents in HepG2 cells 48 h post-transfection. Bars represent the means ± SD of 3 cell fields. CRISPR ALG NPs concentration was 100 µg/mL containing 1 µg CRISPR plasmids. The amounts of DNA plasmids used was 1 μg were combined with 2 μL reagent NanoJuice core reagent and 0.5 μL NanoJuice transfection booster.

**Figure 11 pharmaceuticals-13-00158-f011:**
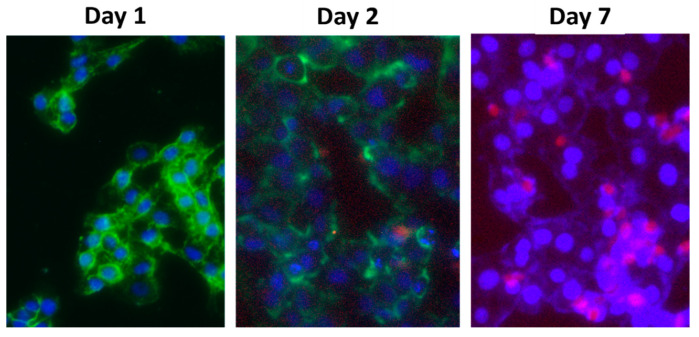
Live cell imaged of HepG2 to visualize their RFP expression 24, 48, and 168 h after transfection of CRISPR plasmids encapsulated into coumarin-6 labeled ALG NPs. Blue is nuclei, red is expressed Cas9-RFP in cytosol, and green is the nanoparticles labeled with coumarin-6. Almost all cells were transfected with the particles, and the majority of particles were localized in cytosol. No expressing of RFP on day 1. On the other hand, the quantity of nanoparticles decreased as it probably degraded or disintegrated and released the plasmid hence the cells start to express Cas9-RFP. Lastly, on day 7, most cells were expressing Cas9-RFP, and almost the cytosol of all cell is absent from the nanoparticles.

**Figure 12 pharmaceuticals-13-00158-f012:**
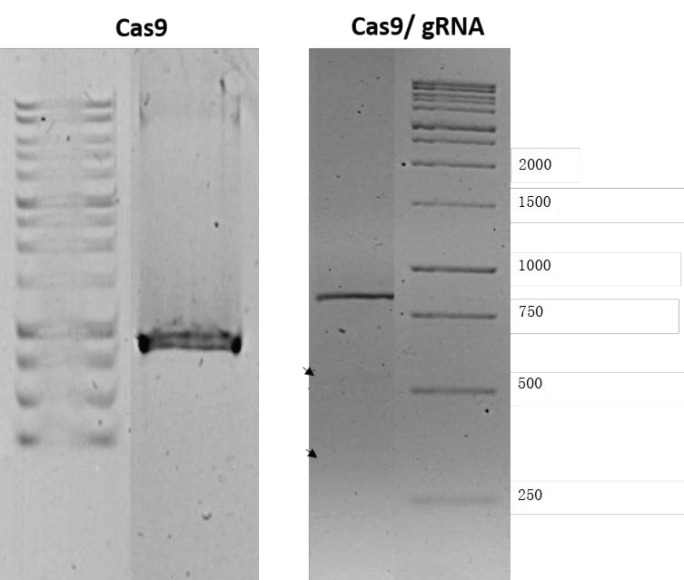
Gel electrophoresis analysis of in vitro ability of CRISPR pDNA delivered by alginate nanoparticles to induce double-strand breaks (DSBs) of targeted DNA. (**left**) PCR amplicon of GFP pDNA delivered to cells treated with Cas 9 only; (**right**) PCR amplicon of EGFP pDNA delivered to cell treated with (CRISPR ALG NPs). Black arrows indicate the new bands results from pDNA cut.

**Table 1 pharmaceuticals-13-00158-t001:** Composition of alginate (ALG) solutions and their physiochemical properties. Superscript letters denote significance, whereby groups that do not share a letter are significantly different *p* < 0.05 (*n* = 3).

Solution	ALG Concentration % *w/v*	Tween 20 % *v/v*	Electrical Conductivity (mS/cm)	Viscosity (mPa S)	Surface Tension (mN/m)	Density (kg/m^3^)	pH	Spray	
								Stability	Mode
S1	0.5	-	1.92 ± 0.09 ^A^	24.45 ± 0.04 ^D^	62.90 ± 0.27 ^A^	1003 ± 0.00 ^C^	6.57 ± 0.010 ^C^	unstable	cone jet
S2	1	-	2.64 ± 0.06 ^B^	59.03 ± 0.05 ^C^	65.02 ± 0.01 ^B^	1004 ± 0.01 ^C^	6.57 ± 0.017 ^C^	unstable	cone jet
S3	1.5	-	3.76 ± 0.05 ^C^	121.26 ± 0.05 ^B^	66.46 ± 0.30 ^C^	1070 ± 0.01 ^B,C^	6.66 ± 0.025 ^B^	unstable	multijet
S4	2	-	4.67 ± 0.02 ^D^	134.85 ± 0.04 ^B^	68.13 ± 0.74 ^D^	1070 ± 0.03 ^B,C^	6.64 ± 0.040 ^B^	unstable	multijet
S5	2.5	-	6.76 ± 0.02 ^E^	138.04 ± 0.05 ^B^	69.99 ± 0.12 ^E^	1090 ± 0.04 ^B^	6.68 ± 0.012 ^A,B^	unstable	multijet
S6	3	-	8.59 ± 0.05 ^F^	356.10 ± 0.08 ^A^	71.30 ± 0.50 ^F^	1170 ± 0.04 ^A^	6.73 ± 0.012 ^A^	unstable	multijet
S7	0.5	1	1.73 ± 0.03 ^G^	30.36 ± 0.04 ^D^	34.86 ±0.16 ^G^	1009 ± 0.00 ^C^	5.47 ± 0.040 ^D^	stable	cone jet
S8	1	1	2.88 ± 0.121 ^H^	59.03 ± 0.05 ^C^	36.29 ±0.16 ^H^	1006 ± 0.00 ^C^	5.47 ± 0.020 ^D^	stable	cone jet

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
