# Peer review of "Electrosprayed Alginate Nanoparticles as CRISPR Plasmid DNA Delivery Carrier: Preparation, Optimization, and Characterization"

_pharmaceuticals, 2020, doi:10.3390/ph13080158_

Round 1

Reviewer 1 Report

In the manuscript by Alallam and colleagues, the authors prepared alginate nanoparticles by electrospray technique for the delivery of plasmid DNA. The authors optimized the preparation process and characterized the obtained nanoparticles, also demonstrating that the activity of encapsulated plasmid DNA is retained. The paper does not bring much novelty, as alginate-based nanocarriers have already been proposed for the delivery of plasmid DNA or other genetic material. Also, optimization of alginate electrospray for the preparation of nanoparticles has been reported as well. The research is of interest to the readers of Pharmaceuticals.

I have the following major concerns:

  • The authors should pay more attention to the reference cited. References 17, 18, 19 and 20 are not appropriate, as they refer to chitosan and not alginate. Also, I was not able to find the article cited as ref. 27.
  • Some procedures are not described. E.g. line 165: how was lyophilization conducted? What were the process parameters? Was a cryoprotectant used? Were all the characterization tests performed on fresh or freeze-dried samples? And the tests on cells? I guess that the lyophilization process will change the characteristics of nanoparticles, e.g. in term of size. Therefore, the authors should clearly define their production process, including the lyophilization, which is an essential step for future commercialization of such nanoparticles and repeat all the characterization and cell tests on the real final product, which is the lyophilized one. It is not acceptable that the sample is freeze-dried only for FTIR analysis, as it seems.
  • I have some concerns about the release studies. Were the sink conditions maintained? Also, how was the calibration curve built?
  • For the uptake experiments, the authors use nanoparticles labelled with coumarin-6. Such nanoparticles were prepared with a slightly different procedure, which may have changed the properties of the final product, e.g. size, morphology, charge. And it is well known that all these affect the nanoparticle uptake by cells. No characterization of these nanoparticles has been reported. Authors must demonstrate that coumarin-6 nanoparticles have the same characteristics as the ones containing plasmid DNA.
  • The manuscript reports a “statistical analysis” paragraph in the materials and methods section. However, it seems that statistical analysis has not been conducted. Authors should present the results with their statistical significance and discuss only significant results. More detail can be found in some comments below.

Minor comments:

  • The abstract should be reduced to 200 words maximum according to the journal guidelines. Please refer to the author’s instruction reported in the template.
  • Line 28: specify the cell line
  • Line 87: “green method” needs to be contextualized. As it is reported, it means nothing.
  • “In more secure and greener manner through avoiding organic solvents in order to achieve more rational, coherent, efficient and cost-effective production procedure.” The authors should demonstrate all these to write them. The term green is also repeated in the conclusion section. Again, this sentence needs to be contextualized. As it is, it is speculative.
  • The author should pay attention to the measuring unit. A concentration CANNOT be defined only with “%”. It has to be specified if it is a w/w, w/v or v/v %. This has to be revised through the entire manuscript.
  • Line 125-126: it is not clear. Add the selected alginate solution between brackets.
  • Equation 1 must be written and not pasted as an image!
  • Line 203-205. Everybody knows how MTT works. There is no need in repeating it. However, I would like to point out that, as clearly specified by the authors, the MTT evaluates cellular metabolic activity. Therefore write “number of viable cells” is incorrect. Cell viability has to be replaced with cell metabolic activity through the entire manuscript.
  • Line 208, 241: specify the concentrations!
  • Line 209-210: was the DMSO added without removing the MTT?
  • Line 212, 553: no! Cell metabolic activity is the correct term.
  • Line 299. It is not so clear that the surface tension increased. Was a statistical analysis conducted? I hope so. Then, statistic must be added to the table.
  • Line 300. This is not true. pH increases!
  • Data in figure 2 must be presented ad shear stress vs shear rate.
  • Line 309-310. Unclear. The authors state that S1 and S2 give a not-stable spray and attribute this to the high surface tension. However, S1 and S2 have not the highest surface tension. As said above, statistic analysis must be done and added to the table to effectively highlight if a statistical difference exists between S1, S2 and the others.
  • Line 330-331: it seems that the conductivity was changed, however! Again, do and add statistical analysis to the table. Then, discuss what is statistically significant.
  • Line 337-344. The data mentioned here have to be fully reported as supplementary material.
  • Line 381-382. Please report the equation model.
  • Line 396-400 and 409-413. References are missing.
  • Line 430-432. Reading these lines, it is clear that the materials and methods section does not fully report how the analysis was conducted. Revise it.
  • Figure 4D. An image at lower magnification and showing more nanoparticles must be provided to demonstrate adequately that aggregates are not present.
  • Line 492: what do essential oils have to do with it? Line 492-495 should be removed.
  • Line 508-510. The data mentioned here have to be fully reported as supplementary material.
  • Figure 9B and 9C: add statistics!

Author Response

Thanks for reviewing and providing invaluable comments to our manuscript. Our response to the comments is attached.

Reviewer 2 Report

The manuscript by Batoul Alallam et al. describes the preparation and characterization of a electrospray-mediated alginate-DNA formulation for delivery of CRISPR genes. The work is interesting and well presented, and the selected methodology is appropriate and well described. I recommend publication after addressing the following minor points:

  1. The language should be revised to correct some typographical errors. As a few examples:
  • Abstract text. “in vivo or ex vivo” should be “in vitro or ex vivo”.
  • Page 2, line 70. “automated to for small…”
  • Page 3, line 88. “Two… has been” instead of “have been”.
  • Page 3, line 91. “fomrulation”
  • Page 3, line 92. “incestigated”
  • Table 1, PH instead of pH.
  • Page 14. PO2 instead of PO2
  • Heading for section 3.2.10. “In virto”
  • Figure 6 caption says “1 and 12 h” of incubation with FBS, but later it says “1 and 24 h”.
  1. Looking at the TEM micrograph presented in Figure 4, it looks like a negatively-stained sample, but the authors state in the Experimental section that no staining was used.
  2. Figure 10 is never mentioned in the text (perhaps it is a typo when Figure 9C is mentioned on page 20, line 644 and it should be Figure 10C?). Figure 11 is mentioned very far from when it is presented.

Author Response

(The authors gave the same response as above.)

Reviewer 3 Report

The manuscript entitled “Electrosprayed Alginate Nanoparticles as CRISPR Plasmid DNA Delivery Carrier: Preparation, Optimization, and Characterization” presents the fabrication of alginate nanoparticles for gene carriers using the electrospray technique. The authors demonstrated this one step green preparation technique requires no additional steps for purification and removal of other ingredients, which would preserve the quality of encapsulated genetic content. Interestingly, the effects of formulation characteristics and processing attributes have been investigated. In addition, these designed nanoparticulate forms are extensively characterized along with biocompatibility and bioactivity. Moreover, the primary results of gene editing capability are also shown as a proof-of-concept. Although the evaluations were done systematically, some irregularities remained to be addressed prior to being publishable on pharmaceuticals.

I suggest adding some recent hot papers in the introduction, International Journal of Nanomedicine 2020:15 675–704, polymer-coated nanoparticles for augmented delivery, Chemical Engineering Journal 383 (2020) 123138.

Moreover, various examples reported studies of gene-encapsulated carriers should be discussed in the introduction to demonstrate why this approach is required.

Provide a schematic to demonstrate the outline of the study.

The authors explained the release of the encapsulated gene but how was it released wasn’t discussed. Moreover, the authors stated…hence alginate hydrogel and its 632 degradation product exhibit ‘proton sponge effect’ that increases the osmotic pressure….” Requiring the demonstration of degradation of the alginate nanoparticles in vitro.

Unify the scale in the ATR-FTIR and merge the Y-axis, some values are edited, better remove the scale

Several grammar and spell errors/irregular and no spacings, for instances, incestegated, ofgene, effeciecntly, . courmarin-9 . Moreover, subscripts and superscripts are completely ignored.

Editing is abysmal, irregular line spacing, and even section headings were mixed with the figure captions.

Author Response

(The authors gave the same response as above.)

Round 2

Reviewer 1 Report

I carefully checked the revised version of the authors. I am satisfied with the changes made. However, supplementary materials cannot be found. They are not at the end of the manuscript, nor online. On the mdpi site it appears "Externally hosted supplementary files", but no link is reported. Therefore I cannot check the data reported in such materials.

Author Response

Thanks for revising our manuscript and checking all our replies.

We apologized of that technical error which made the supplementary materials not uploaded.

Here we attach the supplementary file.

Regards

Round 3

Reviewer 1 Report

The paper can be accepted in present form

Author Response

Thanks very much for approving our revision and for the time spent in making our manuscript better.